# Epinephrine Affects Ribosomes, Cell Division, and Catabolic Processes in *Micrococcus luteus* Skin Strain C01: Revelation of the Conditionally Extensive Hormone Effect Using Orbitrap Mass Spectrometry and Proteomic Analysis

**DOI:** 10.3390/microorganisms11092181

**Published:** 2023-08-29

**Authors:** Andrei V. Gannesen, Rustam H. Ziganshin, Maria A. Ovcharova, Ekaterina D. Nevolina, Alena I. Klimko, Sergey V. Martyanov, Vladimir K. Plakunov

**Affiliations:** 1Federal Research Center “Fundamentals of Biotechnology”, Russian Academy of Sciences, 119071 Moscow, Russia; masha_ovcharova_97@mail.ru (M.A.O.); katia.nevolina@yandex.ru (E.D.N.); alenaklimko221@yandex.ru (A.I.K.); semartyan@inbox.ru (S.V.M.); yavladimir14@gmail.com (V.K.P.); 2Institute of Bioorganic Chemistry, Russian Academy of Sciences, 117997 Moscow, Russia; rustam.ziganshin@gmail.com

**Keywords:** *Micrococcus luteus*, skin microbiota, microbial endocrinology, Orbitrap mass spectrometry, biofilms, planktonic cultures, proteomics, epinephrine, hormones, host–bacteria interactions, catecholamines

## Abstract

**Simple Summary:**

In recent decades, more and more data have been accumulated on the close interaction of the human microbiota with the human organism. Today, we can safely say that human hormones are microbiota regulators that can influence both the behavior of individual species and the balance of microbial communities. However, the mechanisms of such action have been elucidated for an extremely small number of microorganisms. In particular, very little is known about how hormones, especially catecholamines, affect skin Gram-positive microorganisms and their biofilms. In our work, we studied for the first time how epinephrine at a concentration close to normal in blood plasma affects the protein composition of planktonic cultures and biofilms of the skin microorganism *Micrococcus luteus* C01, as well as how the protein composition changes depending on the cultivation time. It was shown that despite the small number of genes with altered expression in the presence of the hormone, the concentration of dozens and hundreds of proteins changes in the presence of epinephrine. The main “targets” of epinephrine are proteins of ribosomes, the tricarboxylic acid cycle, and cell division. Thus, this study opens up great prospects for further work in the field of microbial endocrinology.

**Abstract:**

In the current study, extensive Orbitrap mass spectrometry analysis was conducted for skin strain *Micrococcus luteus* C01 planktonic cultures and biofilms after 24 h and 72 h of incubation either in the presence of epinephrine or without any implementations. The investigation revealed the complex and conditionally extensive effect of epinephrine at concentrations closer to normal blood plasma concentrations on both planktonic cultures and biofilms of skin strain *M. luteus* C01. The concentrations of hundreds of proteins changed during the shift from planktonic growth mode to biofilm and hundreds of proteins were downregulated or upregulated in the presence of epinephrine. Ribosomal, TCA, and cell division proteins appear to be the most altered in their amounts in the presence of the hormone. Potentially, the regulatory mechanism of this process is connected with c-di-GMP and histidine kinases, which were affected by epinephrine in different samples. The phenomenon of epinephrine-based biofilm regulation in *M. luteus* C01 has wide implications for microbial endocrinology and other research areas.

## 1. Introduction

In recent decades, much data has been obtained concerning the interactions between human organisms and human microbiota on a signaling level [1]. As a result, microbial endocrinology [2] has taken shape as a virtually independent scientific area at the interface between microbiology, medicine, and endocrinology. To date, it is known that at least some classes of hormones (i.e., catecholamines) have a crucial impact on bacteria, especially members of the *Enterobacteriaceae* family [3], and on other bacteria genera of different phyla [4,5,6]. Nevertheless, despite the discovery of a complete two-component catecholamine-based signaling system in *Escherichia coli* [3,7], less is known about subsequent biochemical changes in cells and biofilms in the presence of these hormones. Moreover, little is known about how hormones impact the skin microbiota [1], especially regarding Gram-positive species, for instance, micrococci. *Micrococcus luteus* is a saprotrophic Gram-positive bacterium inhabiting different environments such as soil [8], water [9], and industrial waste [10,11]. It is also an essential part of the skin microbial community [12], which is present on the skin surface and inside hair follicles [13]. Not so many works are dedicated to this microorganism, especially its biofilms. As demonstrated in the analysis of the NCBI database, a search conducted on 06 April 2023 for the tag “*Micrococcus luteus*” returned 12,736 results from the literature. Using the tag “*Micrococcus luteus* biofilm” returns only 2264 results. At the same time, the tags “*Staphylococcus aureus*” and “*Staphylococcus aureus* biofilm” return 384,119 and 59,456 results, respectively, and tags “*Pseudomonas aeruginosa*” and “*Pseudomonas aeruginosa* biofilm” return 242,213 and 57,515 results, respectively. This is not surprising considering the fact that *S. aureus* and *P. aeruginosa* are the most dangerous nosocomial pathogens, and *S. aureus* is a bacterium that causes more deaths than AIDS [14]. Therefore, these species are also amongst the most widely used in biofilm models, and *M. luteus* is probably not so attractive due to its relative safety. However, in the last few years, more evidence has appeared of *M. luteus* and the participation of its biofilms in infectious processes [15,16,17,18]. For instance, some data were collected about *M. luteus* causing bloodstream infections [17] and native or prosthetic valve endocarditis [15,16,18], with more published works dedicated to its biofilm [19,20,21]. In addition, recent works have shown that this bacterium is sensitive to epinephrine, and this hormone mostly affects the biofilms of *M. luteus* rather than its planktonic cultures [5,22]. Seven genes were found to be altered in their expression level; however, the proteomic analysis of the *M. luteus* C01 matrix showed dozens of proteins whose concentrations changed in the presence of epinephrine [5,23]. Hence, the effect of the hormone on *M. luteus* seems to be complex and potentially involves some regulatory pathways, as was shown for the *Enterobacteriaceae* family. To take a step toward understanding the mechanism of epinephrine action, the proteomic analysis of whole biofilms and planktonic cultures in the presence of epinephrine was performed in this study to determine whether there are changes in the proteomic profile both of planktonic and biofilm cells and, in addition, how epinephrine affects these processes.

## 2. Materials and Methods

Strains and conditions. *Micrococcus luteus* C01 strain [5,23] was isolated from the skin of a healthy volunteer [24]. The bacterium was stored as previously described [5,22]. Briefly, the biomass was grown in the tubes containing semi-liquid lysogeny broth (LB, Dia-M, Moscow, Russia) agar (0.5%), covered with sterile mineral oil (MendeleevShop, Moscow, Russia) and stored at room temperature (RT). For experiments, the colonies on 1.5% LB agar plates were obtained after the biomass from a single colony was inoculated into a 50 mL conical flask containing 20 mL of LB and incubated at 33 °C and 150 rpm for 24 h. Liquid cultures were then adjusted to OD_540_ = 0.5 with sterile LB and used in experiments.

Cultivation of biofilms and planktonic cultures. Planktonic cultures were obtained in 250 mL conical flasks filled with 100 mL of reinforced clostridial medium (RCM) prepared as previously described (Danilova et al., 2020 [22]). Briefly, 1.5 mL of prepared *M. luteus* culture (OD_540_ = 0.5) was added into medium containing 4.9 nM epinephrine (a concentration close to the physiological blood plasma level and used in previous studies (Gannesen et al., 2021 [5]; Gannesen et al., 2022 [23]). The control flasks lacked additions. Cultures were incubated for 24 h or 72 h at 33 °C and 150 rpm. Then, the biomass was centrifuged at 14,000× *g* (RT) for 20 min and washed twice with a sterile physiological saline (0.9% NaCl). Pellets were harvested in the 1.5 mL microcentrifuge tubes (Eppendorf, Hamburg, Germany) and stored at −80 °C.

Biofilms were obtained in the same way as described elsewhere [23]. Briefly, 20 mL aliquots of the 1.5% RCM agar with or without 4.9 nM epinephrine were administered into 100 mm Petri dishes, cellulose filters were then placed onto the agar, and 0.5 mL of prepared micrococci culture was distributed over the filter surface. Plates were incubated at 33 °C for 24 or 72 h. After the end of incubation, the biomass from two plates was erased and transferred into 1.5 mL microtubes and stored at −80 °C in the same way as the planktonic pellets.

Isolation of proteins. Biomass was resuspended in the sterile physiological saline to obtain the final volume of 10 mL (OD_600_ = 1.0). Then, the suspensions were centrifuged at 4000× *g* (RT) for 20 min to obtain the pellet. After supernatant removal, the pellets were resuspended in 600 µL of lysis buffer. The composition of the buffer was based on the Cold Spring Harbor Protocol (link: http://cshprotocols.cshlp.org/content/2014/9/pdb.rec081273.full (accessed on 20 November 2022)) (with alterations. We used 0.5 mM PMSF (Dia-M, Moscow, Russia) as a protease inhibitor and Triton X-100 (Sigma, Darmstadt, Germany), instead of NP-40, as a detergent. The samples were transferred into the FastPrep tubes with abrasive material (MP Biomedicals, Santa-Ana, CA, USA) and disrupted for 20 s using a FastPrep disintegrator (speed regime 6) for three rounds in succession. The tubes were then centrifuged at 13,000× *g* for 1 min to precipitate the abrasive cell debris. After that, the concentration of proteins was measured via standard Bradford test (Sigma, Darmstadt, Germany). The sample volumes were adjusted to the sample with the lowest protein concentration by addition of lysis buffer. The cell lysate was transferred into 15 mL conical centrifuge tubes (Dia-M, Moscow, Russia) and mixed with 4× volume of acetone (Sigma, Darmstadt, Germany) to precipitate the proteins at the RT for 10 min. After that, tubes were centrifuged at 4000× *g* for 20 min to precipitate the pellet, the liquid was removed, and samples were dried at the RT overnight. 

Proteomics analysis. The proteomic analysis was performed as described elsewhere [23], with changes. First, 400 µL of the biomass lysate was transferred into 15 mL conical centrifuge tubes (Corning, New York, NY, USA). Then, the samples were mixed with a 4× volume (1.6 mL) of acetone (Merck, Darmstadt, Germany) and incubated for 10 min at RT to precipitate the proteins. After precipitation, the samples were centrifuged at RT at 15,000× *g* for 30 min to collect the proteins. The liquid supernatant was removed, and the samples were transferred into sterile microcentrifuge tubes (Eppendorf, Germany) and air-dried at room temperature. Reduction, alkylation, and digestion of the proteins in solution were performed as described previously [25] with minor modifications. Briefly, sodium deoxycholate (SDC) lysis, reduction, and alkylation buffer pH 8.5 contained 100 mM TRIS, 1% (*w*/*v*) SDC, 10 mM TCEP, and 20 mM 2-chloroacetamide were added to a cell sample. The sample was heated for 10 min at 95 °C, and an equal volume of trypsin solution in 100 mM TRIS pH 8.5 was added in a 1:100 (*w*/*w*) ratio. After overnight digestion at 37 °C, peptides were acidified by 1% trifluoroacetic acid (TFA) for SDB-RPS binding, and 20 μg was loaded on three 16-gauge StageTip plugs, equal volume of ethyl acetate was added, and the StageTips were centrifuged at 400× *g*. After washing the StageTips with 100 μL of 1% TFA/ethyl acetate mixture and 100 μL of 0.2% TFA, peptides were eluted with 50 μL of elution solution containing 50% acetonitrile, 45% water, and 5% ammonia. The collected material was vacuum-dried and stored at −80 °C. Before analyses, peptides were dissolved in 2% acetonitrile/0.1% TFA solution and sonicated for 2 min in ultrasonic water bath.

For the MS experiments, samples were loaded to a homemade trap column 20 mm × 0.1 mm, packed with Inertsil ODS3 3 µm sorbent (GL Sciences, Tokyo, Japan), in the loading buffer (2% acetonitrile, 98% H_2_O, 0.1% trifluoroacetic acid) at 10 µL/min flow and separated at RT in a home-packed [26] fused-silica column 300 mm × 0.1 mm packed with Reprosil PUR C18AQ 1.9 (Dr. Maisch, Ammerbuch-Entringen, Germany) into an emitter prepared with P2000 Laser Puller (Sutter, Atlanta, GA, USA). Reverse-phase chromatography was performed with an Ultimate 3000 Nano LC System (ThermoFisher, Waltham, MA, USA), which was coupled to the Q Exactive Plus Orbitrap mass spectrometer (ThermoFisher, USA) via a nanoelectrospray source (ThermoFisher, USA). Peptides were loaded into a loading solution (98% 0.1% (*v*/*v*) formic acid, 2% (*v*/*v*) acetonitrile) and eluted using a linear gradient: 3–35% solution B (0.1% (*v*/*v*) formic acid, 80% (*v*/*v*) acetonitrile) for 105 min; 35–55% B for 18 min, 55–99% B for 0.1 min, 99% B during 10 min, 99–2%B for 0.1 min at a flow rate of 500 nL/min. After each gradient, the column was reequilibrated with solution A (0.1% (*v*/*v*) formic acid, 2% (*v*/*v*) acetonitrile) for 10 min. MS1 parameters were as follows: 70 K resolution, 350–2000 scan range; max injection time, 30 msec; AGC target, 3 × 10^6^. Ions were isolated with 1.4 m/z window, preferred peptide match, and isotope exclusion. Dynamic exclusion was set to 30 s. MS2 fragmentation was carried out in HCD mode at 17.5 K resolution with HCD collision energy of 29%, max injection time of 80 msec, and AGC target of 1 × 10^5^. Other settings: charge exclusion—unassigned, 1, >7.

Raw spectra were processed using MaxQuant 1.6.6.0 (MQ) [27] and Perseus [28]. The data were searched against the *Micrococcus luteus* Uniprot Tremble database, containing canonical and isoform proteins, version from 15 March 2021 (https://www.uniprot.org/ (accessed on 1 March 2023)).

*MaxQuant* search was performed with the default parameter set, including trypsin/p protease specificity, max 2 missed cleavages, Met oxidation, protein N-term acetylation, and NQ deamidation as variable modifications, and carbamidomethyl Cys as a fixed modification, max 5 modifications per peptide, 1% PSM and protein FDR. The following options were turned on: second peptide, maxLFQ, and match between runs. All runs were analyzed as independent experiments and processed in Perseus. 

In Perseus, the protein group results were filtered for contaminants, reverse, and “identified only by site” proteins. Only the proteins with maxLFQ values in at least 3 out of 7 LC–MS runs were used. For them, missing values were imputed from normal distribution with 0.3 intensity distribution sigma width and 1.8 intensity distribution center downshift. 

In silico proteome analysis. The partial visualization of protein clusters was performed using the online resource Protein–Protein Interaction Networks Functional Enrichment Analysis STRING (https://string-db.org/cgi/input?sessionId=bJepuA5XwxF1&input_page_show_search=off (accessed on 1 April 2023)). The online protein analysis was performed using the database UniProt (https://www.uniprot.org/ (accessed on 1 April 2023)) and NCBI protein BLAST (https://blast.ncbi.nlm.nih.gov/Blast.cgi?PAGE=Proteins (accessed on 1 April 2023)).

Statistics. All experiments were conducted in three independent biological repeats. Two-sample *t*-test with permutation-based FDR 5% was applied to search for significantly changing proteins. The *t*-test and the FDR procedure were performed using the Perseus Software (version 2.0.3.1).

## 3. Results

This section may be divided by subheadings. It should provide a concise and precise description of the experimental results, their interpretation, as well as the experimental conclusions that can be drawn.

### 3.1. Proteome Composition of M. luteus C01: General Remarks

In control samples, 1876 proteins were totally identified both in planktonic cultures and biofilms. The proteome in all control samples completely matched between both planktonic cultures and biofilm. However, the addition of 4.9 nM epinephrine led to an increase in protein number to 1949. Moreover, in control samples, 74 proteins were recognized as unique in comparison to epinephrine samples, while in epinephrine samples there were 144 unique proteins. The unique proteins are presented in Appendix A. Here, we use the term “unique” to mark proteins that had no matches between two groups of samples. At the same time, there were no differences between planktonic cultures and biofilms within the control and epinephrine groups. (Appendix A). In epinephrine samples, there were mostly enzymes involved in DNA processing, and also in transport. Despite the data below appearing logical, this protein list becomes much more interesting due to the presence of two ribosomal proteins (D3LKH9—50S ribosomal protein L35 and A0A4P8HIY0—30S ribosomal protein S13). Hence, this fact can be explained according to the two proposed reasons: (i) the hormone administration seems to be the reason for significant changes in translation processes in cells, or (ii) these differences are the result of protein database limitations. However, the ribosome protein composition of our strain *M. luteus* C01 may be altered, and L35 and S13 may potentially be present at non-detectable concentrations in normal conditions (as described in the article of Perry, 2007 [29]. Epinephrine may alter this and “repair ribosomes” in *M. luteus*. However, this may also be explained by the fact that all these so-called “unique” proteins are minor because of the low amounts of unique peptides detected during the mass spectroscopy analysis. Hence, these changes in proteomic profiles between control and epinephrine samples could be a consequence of stochastic errors in the detection of molecules at ultralow concentrations.

### 3.2. How the Proteome of Planktonic Cultures and Biofilms Changes between 24 h and 72 h Samples: Analysis of Differences within the Groups of Control and Epinephrine Samples 

#### 3.2.1. Proteomic Changes in Planktonic Cultures and Biofilms during Long-Term Incubation and in Control and Epinephrine Sample Groups

In this section, the STRING online software (version 11.5) was used to determine and visualize the proteome of *M. luteus* and its clusterization. Despite the shortage of online databases even for reference strain *M. luteus* NCTC 2665, we decided to consider the STRING images as a useful additional tool to visualize the proteome clusterization.

In both planktonic cultures and biofilms, the proteomic profiles changed (i) during long-term incubation (between 24 h and 72 h samples) and (ii) after epinephrine administration (between control and epinephrine samples). The general scheme of proteomic changes is demonstrated in Figure 1. In the beginning, it is necessary to describe how the maturation from 24 h to 72 h changes the proteomic profiles of planktonic cultures and biofilms. First, we analyzed how proteomic profiles changed during the maturation of the cultures and biofilms. In control samples of planktonic cultures (Table 1), there were changes in concentration for only eight proteins—four were more concentrated and four were less concentrated. Interestingly, one of the increased proteins is connected with cell wall synthesis, and a penicillin-binding protein was decreased in concentration in the mature cultures. However, all the changes seem to have a punctated character and in general control planktonic cultures seem to be much more stable in time. Interesting also is the decrease in the penicillin-binding protein at 72 h in the control planktonic cultures, which may potentially lead to higher antibiotic susceptibility. In addition, a component of the cell wall synthesis system was increased in concentration.

Biofilms were more volatile and time-dependent in comparison with planktonic cultures, where there were changes in 429 proteins in 72 h biofilms in comparison to 24 h biofilms in control samples (Appendix A). A total of 330 proteins were increased in concentration in mature 72 h biofilms. Amongst them, the cluster of nucleobase synthesis proteins was increased in concentration. Of the 99 proteins that decreased, there was a large cluster of ribosomal proteins and tricarboxylic acid cycle proteins. Thus, there is generally a decrease in catabolism and protein synthesis in mature biofilms.

In the presence of epinephrine, if we compare the 24 h and 72 h planktonic cultures, there are 50 proteins (33 increased, 17 decreased; Table 2) in 72 h planktonic cultures in the presence of the hormone. Thus, the hormone led to increased proteomic volatility in planktonic cultures. STRING clustering shows that the cluster of Acyl-CoA processing proteins increased in mature cultures, and the same is correct for some Zn-dependent hydrolases (Appendix A), while some ribosomal proteins and proteins involved in cell wall synthesis were decreased in mature cultures. Hence, on the one hand, it seems that in the presence of epinephrine, planktonic cultures tend to slightly increase in resemblance to biofilms, and on the other hand, it seems that epinephrine causes the accelerated “aging” of planktonic cultures.

In biofilms, when we compared the 24 h and 72 h epinephrine samples, it was found that the concentrations of 686 proteins changed, of which 246 were higher and 440 lower in concentration in mature biofilms (Appendix A). The concentrations of several TCA proteins, ribosomal proteins, and Acyl-CoA metabolism proteins were higher in mature 72 h biofilms (Appendix A). Among the suppressed proteins, a large cluster of ribosomal proteins was observed (18 proteins in epinephrine samples instead of 11 in control samples; Appendix A). In addition, some proteins of the nucleobase synthesis process and tRNA-AA assembly were lower in concentration. Hence, the synthesis of nucleic acids and proteins was impaired to an even greater extent in mature biofilms in the presence of epinephrine, which also suggests there is acceleration of maturation and “aging” in the presence of the hormone.

#### 3.2.2. Proteomic Changes in Planktonic Cultures in Comparison with Biofilms in Control and Epinephrine Sample Groups

As has been established for decades, the biofilm phenotype appears as a consequence of global gene expression shift, which results in changes in the concentrations of dozens to hundreds of proteins. When we compared control samples of 24 h planktonic cultures and biofilms, we found changes in the concentrations of 524 proteins in biofilms. In that protein set, 281 were higher and 243 lower in concentration (Appendix A). In the case of the proteins at higher concentration, some are involved in DNA and RNA polymerization processes, there are two enzymes of GMP/GTP synthesis, four ribosomal proteins, four proteins involved in the amino acid synthesis, and also FtsZ (Appendix A). Hence, it seems that, at least partially, the cells in biofilms are more metabolically active, produce more proteins, and potentially have accelerated cell division. Histidine kinase A0A5F0IAR6 was among the proteins stimulated in biofilms. Among the decreased proteins, there was a large cluster of Acyl-CoA synthesis proteins, thiamine synthesis proteins, some enzymes of amino acids synthesis, thioredoxin and thioredoxin reductases, and two proteins of the OmpR two-component signaling system (Figure 1 and Appendix A). The last two groups, and also the histidine kinase acc. # A0A031GZ94 of reduced concentration (Appendix A), seem to be components of a whole signaling system that may represent a potential signal system—of which there is at least one—responding to epinephrine [30].

Comparing the mature 72 h planktonic cultures and biofilms without epinephrine implementation, we found that in biofilms, there were increased concentrations of some enzymes of Acyl-CoA processing and potentially lipid synthesis (Figure 1, Appendix A), some thiamine synthesis enzymes, some TCA and glycolysis enzymes, nucleobase synthesis proteins, and two proteins of the two-component signal system of the OmpR family. However, a large cluster of ribosomal proteins, some enzymes of the TCA cycle, some proteins of cell wall synthesis, and especially FtsZ were decreased in concentration (Figure 1 and Appendix A). Hence, 72 h biofilms were much less metabolically active than planktonic cultures. Interestingly, concentrations of proteins of the two-component OmpR signaling system were higher in mature biofilms than in planktonic cultures. Hence, it could be suggested that the OmpR family two-component system may be involved in mature biofilm metabolism and, potentially, in biofilm dispersion processes. 

When we compared the 24 h planktonic cultures and biofilms in the presence of epinephrine, we found that the changes were closer to the changes in 72 h planktonic cultures and biofilms in control samples. Here, there were partial increases in the cluster of thiamine synthesis and Acyl-CoA processing proteins (Figure 1 and Appendix A), and FtsZ, some proteins of the TCA cycle, and another cluster of Acyl-CoA processing proteins were lower in concentration (Appendix A). Moreover, some proteins of aromatic amino acids synthesis were lower in concentration in 24 h biofilms in the presence of epinephrine in comparison with planktonic cultures. Histidine kinase A0A031GKK8 was also suppressed by epinephrine. Hence, we could suggest that this is further support for the hypothesis of the partial “aging” effect of epinephrine, at least regarding some aspects. At the same time, there were no shifts in the ribosomal proteins. In general, the concentrations of 168 proteins changed in 24 h biofilms in the presence of epinephrine: 93 were higher and 75 lower (Appendix A).

When we studied 72 h planktonic cultures and biofilms in presence of the hormone, we found that, first, the changes in ribosomal proteins were not as dramatic as in control samples, and some ribosomal proteins were higher in concentration in biofilms (Figure 1, Appendix A). Moreover, a cluster of Acyl-CoA (and potentially lipids) synthesis proteins and TCA cycle proteins were in higher concentrations in biofilms in comparison with the planktonic cultures. Concerning the proteins lower in concentration, there were decreases in some ribosomal proteins, proteins of DNA replication and tRNA-AA assembly, TCA cycle proteins, and Acyl-CoA processing and cell division proteins, including the FtsZ (Figure 1 and Appendix A). Hence, the addition of epinephrine to the medium led to an attenuation in the decrease in ribosomes, shifts in Acyl-CoA (and potentially lipids) synthesis and in the TCA cycle. However, as in the control samples, cell division seems to be impaired in biofilms in comparison with planktonic cultures after 72 h of incubation with the hormone. In general, the concentrations of 585 proteins changed in 72 h biofilms in the presence of epinephrine: 295 were higher and 290 lower (Appendix A).

### 3.3. The Comparison of Control and Epinephrine Samples of Planktonic Cultures and Biofilms: Effect of the Hormone in Planktonic Culture and Biofilm Sample Groups

To analyze the possible effect of epinephrine on planktonic cultures and biofilms in comparison with control samples, we analyzed both 24 h and 72 h samples.

#### 3.3.1. The Effect of Epinephrine on *M. luteus* C01 Planktonic Cultures

Both in 24 h and 72 h samples, the addition of epinephrine caused an increase in the amount of some ribosomal and TCA cycle proteins (Figure 1 and Appendix A). Moreover, in 24 h samples in the presence of epinephrine, there were increases in the concentration of some cell wall and DNA synthesis proteins and also the elongation factor Ef-Tu (Figure 1 and Appendix A). In 72 h samples, in the presence of the hormone, there were also increases in some proteins of Acyl-CoA synthesis and the TCA cycle cluster (Figure 1 and Appendix A). In turn, in both 24 h and 72 h planktonic cultures, there was a decrease in some nucleobase synthesis proteins (Figure 1 and Appendix A). Hence, epinephrine also potentially acts as a nucleobase synthesis inhibitor in planktonic cells of *M. luteus* C01. In 72 h planktonic cultures, epinephrine also decreased the amounts of some proteins involved in cell division (including FtsZ), tRNA-AA assembly, translation, amino acids synthesis, stress response, ribosomes, and nucleic acid polymerization (Figure 1, Appendix A). Hence, in mature planktonic cultures, the addition of epinephrine shifts the balance in ribosomal proteins and potentially impairs nucleic acid production and cell division, which supports the hypothesis of the “aging” effect of the hormone. In general, after 24 h of incubation, the concentrations of 220 proteins changed in planktonic cultures in the presence of epinephrine: 87 were higher and 133 lower (Appendix A). After 72 h of incubation, the concentrations of 332 proteins changed: 100 were higher and 232 lower (Appendix A).

#### 3.3.2. The Effect of Epinephrine on *M. luteus* C01 Biofilms

The investigation of biofilms showed that the hormone has a greater impact on biofilms than on planktonic cultures. Generally, after 24 h of incubation, the concentrations of 462 proteins changed (Appendix A): 225 were higher and 237 lower. After 72 h of incubation, the concentrations of 276 proteins were higher and 830 lower, corresponding to a total of 1006 proteins (Appendix A).

The first phenomenon that should be mentioned is that both after 24 h and 72 h of incubation, in comparison with control samples, the hormone increased the large cluster of ribosomal proteins and some TCA cycle and electron transport chain (ETC) proteins (Figure 1 and Appendix A). Moreover, in 24 h samples, epinephrine stimulated the elongation factor Ef-Tu in the same way as in planktonic cultures (Figure 1 and Appendix A) and some proteins of nucleobase synthesis and acyl residues processing. After 72 h of incubation, there were increased concentrations of some chaperones and enzymes involved in ATP synthesis (Figure 1 and Appendix A). All those facts suggest the general acceleration of cell anabolism in biofilms by the hormone in comparison with the control samples. Concerning the proteins that became lower in concentration, epinephrine suppresses FtsZ after 24 h of cultivation (Figure 1 and Appendix A). Histidine kinase sensor A0A2N6RI38 was also among the suppressed proteins. After 72 h of cultivation, in addition to FtsZ, some other cell wall synthesis proteins, proteins of nucleobase synthesis, and some proteins of the two-component OmpR family signal system were decreased in concentration (Figure 1 and Appendix A). The histidine kinase A0A410XQU6 was also among the suppressed proteins. Hence, on the one hand, there is a pronounced acceleration of cell anabolism and synthesis of proteins, nucleic acids, and ATP. This is a controversial “aging” effect, which seems to be opposite to that in other cases in this study. On the other hand, there was a decrease in more than 800 proteins in 72 h biofilms. Moreover, it seems that epinephrine suppresses FtsZ and cell division. Thus, the increase in ribosomal proteins may not be directly connected with actual cell proliferation because of the potential to block translation. 

### 3.4. Comparison with the Biofilm Matrix Proteome 

In addition, we analyzed data obtained in previous work [23] and built a general clusterization using the STRING online service. The biofilm matrix of *M. luteus* contains more than 700 proteins, with many ribosomal proteins, ETC proteins, cell division proteins (including FtsZ), transport proteins, and even signaling proteins (Figure 2). Hence, there is the question of whether it is possible that epinephrine causes mostly changes in the matrix but not in the cellular proteome.

We analyzed whether there were changes in the matrix proteome in the presence of epinephrine (Figure 3). We found that several ribosomal proteins showed higher concentrations at 24 h in the biofilm matrix after the administration of the hormone (Figure 3A). Additionally, FtsZ and some cell wall and division proteins also showed higher concentrations in the matrix following epinephrine administration. 

After 72 h of incubation, only four ribosomal/ribosome-associated proteins showed higher concentrations in the biofilm matrix after epinephrine addition (Figure 3B). However, there were several TCA/ETC proteins and also two cell division proteins. Hence, we suggest that, because of a “leak” of some ribosomal proteins and some of the FtsZ pool from the cells to the matrix, it is possible that, due to an increase in the concentrations of some ribosomal proteins in the whole biofilms, an imbalance inside cells could have been induced, leading to the interruption of protein synthesis. This could be one of the reasons there was a decrease in so many of the proteins in the biofilms in the presence of epinephrine after 72 h of incubation. Similar results were observed for FtsZ. The increase in its concentration in the matrix caused by the hormone and the decrease within the total biofilm could be explained by a “leak” of FtsZ from cells to the matrix.

We also found that about a hundred (specifically, 124) of the proteins in the matrix were “unique” and were not found in the total proteome of the biofilms studied in this work. They are presented in Figure 4 and in Appendix A. Amongst them, there were some cell division proteins, some transport proteins, and others. It would be illogical for the matrix to contain proteins that were absent in the whole biofilm; this apparent illogical result could be explained by methodological issues. When we worked with the biofilm matrix as described previously [23], we collected a very large amount of the biofilm biomass (up to several grams per repeat). In the present study, we destroyed a very small amount of biomass (several milligrams). Hence, it is possible that, due to the different concentrations of such proteins in the matrix and in the cells, and due to the different biomass amounts, the concentrations of those proteins in the present study were too low for them to be well recognized using mass spectrometry. In other words, it is possible that peptides present at ultralow concentrations were stochastically missed altogether during the mass spectrometry detection. Additionally, there may be database errors or shortened protein names that made some of the analysis inaccurate. 

## 4. Discussion

To date, the accumulated evidence demonstrates that the skin microbial community is potentially also regulated by human humoral systems [1,31]. Nevertheless, some critical methodological and generally scientific points await resolving in studying how the human skin microbiota reacts in the presence of hormones: (i) How exactly are hormones present in human skin and where are they concentrated? Some studies connected mostly with the administration of different steroid-based contraceptives revealed the concentration of female steroids in biopsy samples of lower urogenital tracts [32]. As was shown, steroid (i.e., estradiol) concentrations were normally lower in tissues than in blood plasma. However, no information was found about concentrations of any hormones in the skin. Hence, despite the case of female steroids, the only way that seems to be appropriate is to rely on blood plasma levels. (ii) How do the internal effects of microbes on each other inside the complex biofilm community actually interfere with external molecular impacts? On the one hand, in vitro and in vivo models based on monospecies cultures and biofilms have been established for decades and provide an abundance of important data. On the other hand, there are several lines of evidence of interference due to (a) different active compounds being administered simultaneously [33,34]; (b) the effects of microbes on each other [19,35,36]; (c) modulation of an active compound effect on the first microbe in the presence of a second microbe in a dual-species community [37,38,39]. Hence, the presence of a counterpart in the simplest dual community may significantly change the effect of a hormone or any other active compound on a microbe. What if there are dozens or hundreds of species and dozens of different internal and external regulatory molecules interacting simultaneously? That is the question for future studies (potentially involving AI). Finally, what benefits can we derive from such investigation? Is it possible to apply hormones in humans as microbial regulators in industry, for instance in cosmetology or in pharmacy? Despite some existing data [40,41], the problem seems to be still far from being resolved.

The measurement of gene expression via modern high-throughput methods such as RNAseq is considered a rather reliable approach to determine, at least partially, distinct molecular targets for active compounds or the physiological response of cells to environmental shifts. However, as was established both for eukaryotic [42,43] and prokaryotic [44] organisms, gene expression profiles frequently constrain the proteomic profile of cells. Protein synthesis depends on numerous factors, including ribosome numbers, availability of nutrients, availability of amino acids and amino acid synthesis, and cell energy metabolism machinery. Moreover, it may also differ based on the dependence of cell volume and cell division processes due to the different spatial distribution of different molecular clusters in the cytoplasm [45]. Hence, taking this into consideration, it is not surprising that despite epinephrine significantly affecting the expression of at least seven genes in *M. luteus* C01 as was previously shown [5], the proteomic profile of both planktonic cultures and biofilms does not actually match the gene expression profiles. Dozens and hundreds of proteins changed in concentration during maturation of the cultures and biofilms, hundreds of proteins are different in planktonic cultures and biofilms, and hundreds between control and epinephrine samples. It is also interesting that proteins encoded by genes revealed in the previous study are not actually downregulated or upregulated by epinephrine despite their significantly altered gene expression levels being established. The study of *M. luteus* C01 biofilm matrix composition [23] also revealed similarly extensive changes in the proteomic profiles of matrix composition caused by 4.9 nM of epinephrine in the medium. The dramatic shift in proteomic profiles between planktonic cultures and biofilms is also caused by microenvironmental interactions and the availability of nutrients for cells.

In comparison with control samples, hormone addition leads to increased numbers of ribosomal proteins, TCA and ETC proteins, and other enzymes responsible for the synthesis of polymeric compounds. At the same time, the hormone potentially impairs cell division processes in biofilms, which is reflected in the reduction in the amount of FtsZ and some other cell division and cell wall synthesis proteins. Altogether, these facts directly match some recently described points [44]. Carbon catabolism, ribosomes, and cell division were dramatically changed in the presence of the hormone; thus, the posttranslational effects of epinephrine seem to be multitarget and complex. Moreover, the increase in protein concentrations may be transferred in the biofilm matrix, which may be the reason for the decrease in more than 800 proteins in mature 72 h biofilms, in comparison with the control, due to the misbalance of ribosome composition. On the other hand, after the comparison of changes in 24 h and 72 h samples with epinephrine, with the same changes in control samples, it is clear that the hormone stimulates the decrease in ribosomal proteins and, partially, cell wall synthesis in planktonic cultures and attenuates the inhibition of ribosomes in biofilms. There is also stimulation of TCA in biofilms. Thus, epinephrine generally causes planktonic cultures to be more sensitive to maturation, which results in concentration changes in an increased number of proteins after 72 h of incubation, which also makes planktonic cultures more “biofilm-like”. This absolutely matches the data about biofilms stimulated by epinephrine that we previously obtained [44]. However, this stimulation seems to be mostly in the first 24 h of incubation. Ribosomes, FtsZ, and Acyl-CoA metabolism seem to be the preferred targets of epinephrine. The question, however, is what is the global regulatory mechanism of this effect of epinephrine? Does it exist or is the effect determined by simultaneous decentralized multitarget impacts that interfere with each other to form the complex result? Because we frequently found that in the presence of epinephrine, especially in biofilms, there are some histidine kinases or histidine kinases sensors (for instance, OmpR family proteins of two-component signal systems) among the affected proteins (more often suppressed than stimulated), and given there is a global regulatory mechanism at the base of epinephrine effect, we wonder if it is somehow connected with the regulatory pathways where OmpR histidine kinases play a key role, and where c-di-GMP may be a potential regulator involved in the process as was previously described [45]. Moreover, in some cases, for instance in the 72 h epinephrine biofilms in comparison with control biofilms, the GMP synthetase A0A653T0K5 was stimulated. However, in 72 h control biofilms, the GMP synthase A0A6N4F9N7 was in a lower concentration in comparison with the control planktonic cultures. With epinephrine addition, the GMP synthase C5CBZ4 was decreased in 24 h biofilms in comparison with 24 h planktonic cultures. Thus, all of this may be links in a chain.

Hence, clarification of the actual mechanism of epinephrine action on the actinobacterium *M. luteus* C01 awaits further studies. Furthermore, several questions are also as relevant as deciphering the mechanism. The first is whether the response of *M. luteus* C01 to epinephrine is universal for all the *M. luteus* strains, including non-skin strains from soil, waste, and water. The second is whether the molecule of epinephrine is unique and specific in triggering such a response in *M. luteus* and whether there is any universality in action for molecules chemically similar to epinephrine. The third is whether the potential universal mechanism underlying the response is dependent on epinephrine concentration. The third question may be answered positively; however, consensus is needed. Finally, the fourth question is whether this mechanism is dependent on the presence of other microorganisms in the community, as was shown for A-type and C-type natriuretic peptides, for instance [37,38,39].

## 5. Conclusions

This manuscript describes the first deep investigation into the whole proteome of skin actinobacterium *M. luteus* strain C01, its planktonic cultures and biofilm, and its response to epinephrine. Despite the current lack of certainty in deciphering the mechanism and the apparent complexity of the overall whole picture, the data obtained in our research demonstrate the globality and extensivity of epinephrine action on this microorganism. New evidence was obtained for how important it is to study the effects of hormones on all representatives of the human microbiota. We also gained additional support for the importance and universality of the area of microbial endocrinology, and our results will certainly help us in further understanding the mechanisms of action of epinephrine on not only micrococci but also other bacterial and non-bacterial species inhabiting humans and, particularly, our skin. This has wide implications for both fundamental science and research on new potential applications in the future.

## Figures and Tables

**Figure 1 microorganisms-11-02181-f001:**
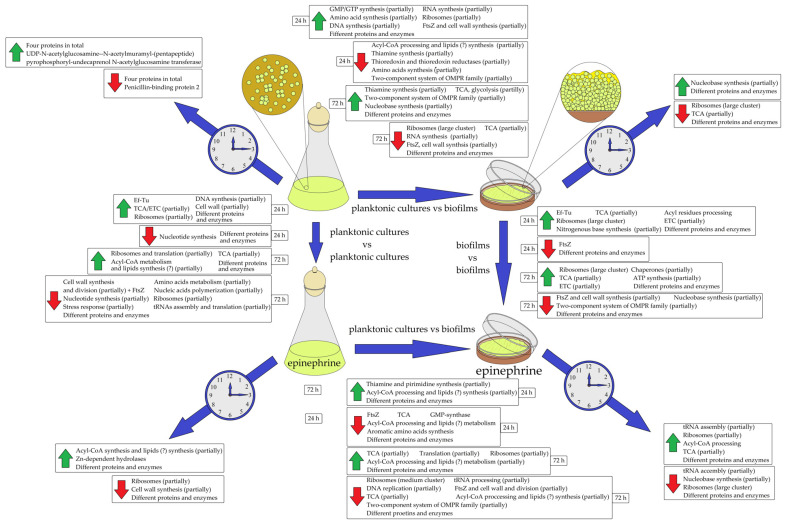
General scheme of proteome alterations in *M. luteus* C01 planktonic cultures and biofilms during maturation of the cultures and biofilms and in the presence of epinephrine. Blue arrows indicate comparison, green arrows indicate an increase, and red arrows a decrease in the protein concentration of the sample to which the blue arrow is pointing. Clock arrows indicate the comparison between 24 h and 72 h samples.

**Figure 2 microorganisms-11-02181-f002:**
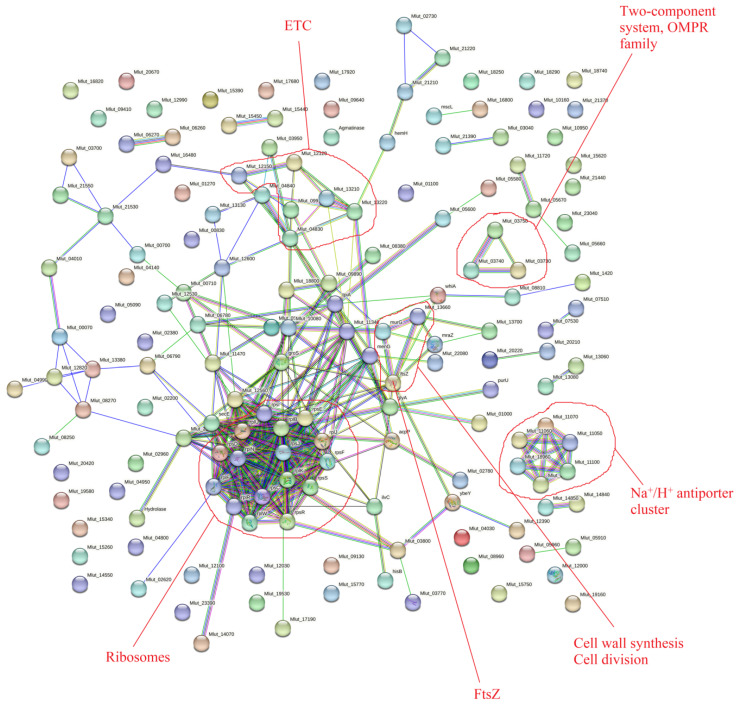
General partial STRING visualization of *M. luteus* C01 biofilm matrix proteome. The data were obtained from Gannesen et al., 2022 [23].

**Figure 3 microorganisms-11-02181-f003:**
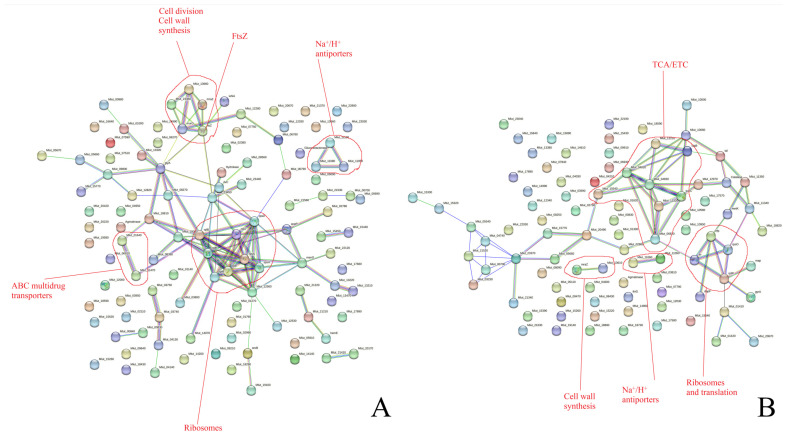
Proteins that showed higher concentrations in the *M. luteus* C01 biofilm matrix in the presence of epinephrine after 24 h (**A**) and 72 h (**B**) of incubation. The data were obtained from Gannesen et al., 2022 [23].

**Figure 4 microorganisms-11-02181-f004:**
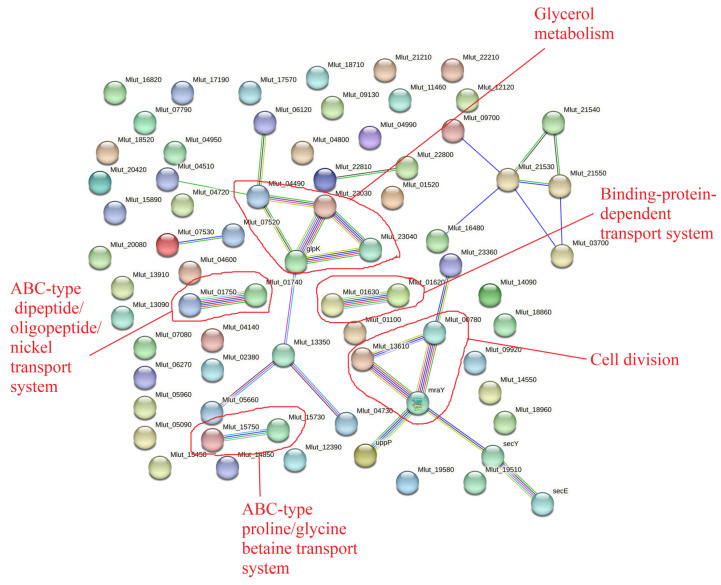
“Unique” proteins in the *M. luteus* C01 biofilm matrix. The data were obtained from Gannesen et al., 2022 [23].

**Table 1 microorganisms-11-02181-t001:** Proteomic changes in *M. luteus* C01 planktonic cultures (controls) from 24 h to 72 h of incubation. Regarding protein concentration in 72 h samples, a green label indicates an increase while a red label indicates a decrease.

Fold Change	Log Student’s *t*-Test *p*-Value	Student’s *t*-Test Significant	Student’s *t*-Test q-Value	Andromeda Score	Peptides	Unique Peptides	Majority Protein IDs	Protein Name
0.046	0.0000575	+	0.000	167.950	11	11	A0A4Y8PNI0	Acetohydroxy acid synthase small subunit
0.367	0.0000009	+	0.000	35.724	6	5	A0A5F0I5D1	UDP-N-acetylglucosamine–N-acetylmuramyl-(pentapeptide) pyrophosphoryl-undecaprenol N-acetylglucosamine transferase
0.611	0.0000143	+	0.000	70.148	12	8	C5CA30	DASS family sodium-coupled anion symporter
0.677	0.0001048	+	0.040	202.960	9	9	D3LL93	PKD domain-containing protein luteus
1.765	0.0000462	+	0.000	190.430	8	6	A0A562G5G6	Enoyl-CoA hydratase/carnithine racemase luteus
2.069	0.0000536	+	0.000	138.000	7	1	A0A5F0IA97	tRNA(Ile)-lysidine synthase
12.426	0.0000283	+	0.000	100.360	8	2	A0A562FUE9	Penicillin-binding protein 2
31.097	0.0000167	+	0.000	3.151	17	0	A0A6N4C4E7	Adenylosuccinate lyase

**Table 2 microorganisms-11-02181-t002:** Proteomic changes in *M. luteus* C01 planktonic cultures (epinephrine) from 24 h to 72 h of incubation. Regarding protein concentration in 72 h samples, a green label indicates an increase while red label indicates a decrease.

Fold Change	Student’s *t*-Test *p*-Value	Student’s *t*-Test Significant	Student’s *t*-Test q-Value	Andromeda Score	Peptides	Unique Peptides	Majority Protein IDs	Protein Name
0.069341	0.000904343	+	0.055778	4.3162	38	1	A0A5F0IA49	AMP-dependent synthetase
0.078911	0.000186892	+	0.031636	53.67	30	0	D3LKU1	Gamma-glutamyl phosphate reductase
0.080156	6.10057 × 10^−5^	+	0.044	10.718	12	0	A0A378NGS6	NUDIX domain
0.106835	5.3882 × 10^−5^	+	0.055	323.31	34	1	A0A6N4C449	Methylcrotonoyl-CoA carboxylase
0.130453	0.000716674	+	0.051625	11.453	19	0	A0A6N4F6V6	Sulfurtransferase
0.142135	0.000366145	+	0.032952	323.31	16	8	A0A5F0IA94	3-hydroxyisobutyrate dehydrogenase
0.145986	0.000147486	+	0.0348	301.7	28	1	A0A5F0I7U0	Acetyl/propionyl-CoA carboxylase subunit alpha
0.154537	0.000474668	+	0.054	102.96	17	5	A0A509Y5D9	Long-chain fatty acid–CoA ligase
0.161394	3.96082 × 10^−5^	+	0.073333	323.31	39	0	C5C8P4	Acyl-CoA synthetase/AMP-acid ligase
0.180196	0.000301722	+	0.033111	128.37	13	13	A0A1M7AJD2	Glutaryl-CoA dehydrogenase
0.191962	0.000600747	+	0.054571	14.963	18	1	A0A6N4FA22	Lysine–tRNA ligase
0.195861	2.78807 × 10^−6^	+	0.096	323.31	17	7	A0A653J4T6	Acyl-CoA dehydrogenase
0.207599	0.000241677	+	0.03725	323.31	13	7	A0A031GG35	GroES-like protein
0.220707	0.00020476	+	0.029	205.13	18	0	A0A2N6RHX4	Acyl-CoA synthetase
0.23262	0.000575452	+	0.056593	323.31	20	1	A0A4Y8PJX4	Acetyl-CoA C-acetyltransferase
0.245841	0.000497764	+	0.0568	323.31	27	2	A0A5E8QG57	Enoyl-CoA hydratase/isomerase family protein
0.250082	0.000268107	+	0.035059	323.31	23	23	A0A1M7AJR6	KR domain-containing protein
0.273894	0.0009515	+	0.0502	237.02	21	19	A0A509Y4M5	Enoyl-CoA hydratase
0.282804	6.30992 × 10^−5^	+	0.031429	115.88	22	0	A0A5F0IB54	3-hydroxyacyl-CoA dehydrogenase
0.310135	0.000949188	+	0.051487	51.029	6	6	A0A5F0IBJ1	N-acetyltransferase
0.347456	0.001145003	+	0.05234	323.31	30	3	A0A4Y8ZJ73	Aldehyde dehydrogenase family protein
0.371518	0.00068292	+	0.055067	64.194	12	3	A0A5F0IAP2	Acyl-CoA dehydrogenase OX = 1391911
0.417764	0.000218222	+	0.036308	296.24	20	20	A0A031IKQ5	Alpha-ketoacid dehydrogenase subunit beta
0.422328	0.000110642	+	0.024444	9.4998	50	0	A0A4U1LK82	Aldehyde dehydrogenase family protein
0.451917	0.000898155	+	0.057371	264.55	13	0	D3LRU4	GroES-like protein SK58
0.483193	0.000621288	+	0.05269	98.964	10	10	A0A031G8N2	ATP-grasp superfamily enzyme
0.50694	0.00098185	+	0.048976	161.74	8	1	C5C833	Predicted Zn-dependent hydrolase of beta-lactamase
0.525612	0.001371883	+	0.0492	323.31	17	1	C5CBR0	Iron-regulated ABC transporter ATPase subunit SufC
0.543036	0.001042716	+	0.048696	277.87	20	20	A0A2N6RPP3	Pyruvate dehydrogenase (Acetyl-transferring) E1 component subunit alpha
0.593636	0.000842068	+	0.055882	323.31	51	37	A0A031GU24	Acyl-CoA synthetase (NDP forming)
0.730556	0.001022347	+	0.048091	187.01	18	16	A0A378NRK2	4-hydroxy-3-methylbut-2-enyl diphosphate reductase
0.787446	7.26814 × 10^−5^	+	0.0275	323.31	34	1	A0A562FU00	Gamma-glutamyl phosphate reductase
0.863303	0.000219033	+	0.033714	78.573	11	1	A0A378NK55	Farnesyl diphosphate synthase
1.354815	0.000808918	+	0.057576	259.17	21	21	A0A5F0I7M5	4-aminobutyrate–2-oxoglutarate transaminase
1.395987	0.00102733	+	0.047022	27.183	4	4	A0A5E8QGG7	Vitamin K epoxide reductase
1.463339	0.000229784	+	0.031467	21.431	6	6	A0A378NL43	HTH-type transcriptional regulator gltC
1.510947	0.000996293	+	0.046698	125.83	12	12	A0A031IW78	Glutamate racemase
1.581977	0.000349861	+	0.036421	278.5	25	25	C5CC59	50S ribosomal protein L2
1.609	0.001319842	+	0.05125	81.518	8	8	D3LR26	30S ribosomal protein S19
1.671859	0.000985313	+	0.04781	149.72	13	13	A0A562FW33	5-oxoprolinase subunit A
1.777925	0.000927898	+	0.05427	59.002	10	10	A0A6N4FDF3	Pantothenate kinase
1.842939	0.00052194	+	0.054615	323.31	93	0	A0A5F0I860	DNA-directed RNA polymerase subunit beta
1.895257	0.000947341	+	0.052842	196.1	19	19	A0A562FSS5	50S ribosomal protein L3
2.401196	1.73241 × 10^−5^	+	0.11	5.7013	2	2	A0A2N6RLH0	Methyl viologen resistance protein SmvA
2.782218	0.000355313	+	0.0346	81.878	6	6	A0A2N6RLI0	TetR/AcrR family transcriptional regulator
2.947873	0.000401354	+	0.04087	75.154	23	19	A0A378NLV1	Linear gramicidin synthase subunit D
3.011034	0.001340022	+	0.050204	23.859	2	1	D3LMW7	Sortase family protein
3.126505	0.00039416	+	0.037091	106.79	10	9	A0A132HXJ2	Siderophore-interacting protein
5.335081	0.000709395	+	0.05329	81.796	8	2	A0A562FUE9	PKD domain-containing protein
14.36734	6.13161 × 10^−5^	+	0.036667	10.524	19	0	A0A653IYK6	UDP-N-acetylmuramoylalanine–D-glutamate ligase

## Data Availability

The mass spectrometry proteomics data have been deposited to the ProteomeXchange Consortium via the PRIDE [46] partner repository with the dataset identifier PXD041922.

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
