# Peer review of "Epinephrine Affects Ribosomes, Cell Division, and Catabolic Processes in Micrococcus luteus Skin Strain C01: Revelation of the Conditionally Extensive Hormone Effect Using Orbitrap Mass Spectrometry and Proteomic Analysis"

_microorganisms, 2023, doi:10.3390/microorganisms11092181_

Round 1
Reviewer 1 Report
In the manuscript entitled ‘Epinephrine affects ribosomes, cell division and catabolic processes in Micrococcus luteus skin strain C01: revelation of the conditionally extensive hormone effect using Orbitrap mass spectrometry and proteomic analysis’. Andrei V. Gannesen et al. carried out proteomics analysis on strain Micrococcus luteus C01 planktonic cultures and biofilms treated with epinephrine. They revealed the effects of epinephrine on proteome changes in both planktonic cultures and biofilms of skin strain M. luteus C01. They found that epinephrine led to changes in proteins associated with ribosomes, TCA and cell division, and they suggested that regulatory mechanism was connected with c-di-GMP and histidine kinases. Although the study was interesting, the results were quite primitive. More experiments were needed to support their conclusions.
1. Authors need to provide detailed information on determine the fold change presented in Table 1. It is also important to validate some differentially expressed proteins by western blotting or targeted proteomic analysis.
2. Authors should demonstrate that c-di-GMP treatment can cause similar effects as epinephrine does. Can they determine which histidine kinases are involved in epinephrine mediated processes?
3. Are there differences caused by epinephrine between planktonic cultures and biofilms.
Minor editing of English is needed.
Author Response
Reviewer 1
In the manuscript entitled ‘Epinephrine affects ribosomes, cell division and catabolic processes in Micrococcus luteus skin strain C01: revelation of the conditionally extensive hormone effect using Orbitrap mass spectrometry and proteomic analysis’. Andrei V. Gannesen et al. carried out proteomics analysis on strain Micrococcus luteus C01 planktonic cultures and biofilms treated with epinephrine. They revealed the effects of epinephrine on proteome changes in both planktonic cultures and biofilms of skin strain M. luteus C01. They found that epinephrine led to changes in proteins associated with ribosomes, TCA and cell division, and they suggested that regulatory mechanism was connected with c-di-GMP and histidine kinases. Although the study was interesting, the results were quite primitive. More experiments were needed to support their conclusions.
- Authors need to provide detailed information on determine the fold change presented in Table 1. It is also important to validate some differentially expressed proteins by western blotting or targeted proteomic analysis.
Answer
Dear Reviewer,
Thank you for the remark. We completely modified the tables with quantitative data and now all the tables contain p-values, q-values and other parameters calculated in the article.
Authors are grateful to the Reviewer for the valuable remarks. Indeed, the data provided in the current manuscript are rather preliminary and need to be extended and validated by other approaches. And we will prolong our investigation with no doubts. However, we suppose that on the current step the data presented in the article are sufficient to be published in the article. Here we will explain our point of view.
The first and the main point is that we used high resolution Orbitrap method for three independent biological repeats in each sample. Thus, we investigated in total 24 samples, which represent good statistical basis of the study. Also, because of Orbitrap is widely used in targeted proteomics studies, an additional use of this very effective MS approach seem to be excess for the investigation.
Next, the second point is that too many proteins were downregulated/upregulated in presence of epinephrine in M. luteus both in planktonic cells and in biofilms. Especially in comparison with the previous data of gene expression it was a rather “surprising” (7 genes revealed via RNAseq+qPCR and dozens and even hundreds of proteins). Hence, it is not so clear what exactly is needed to be validated on this step. What group/groups of proteins should be a primal “target/targets”. And it is not clear what to do with a vast amount of proteins marked as “unidentified” in present databases and have no any closer proteins with studied functions. Even if some of such unidentified proteins are validated, we will still have no idea what exactly functions they do perform in cells/biofilms. So, there is another study raising to be needful: to extract all these proteins and try to found their structure and functions. However, no doubts, these investigations are needful for the future research.
Concerning the WB, it may be useful for less extensive proteomic research as for blood plasma proteins or some very narrow investigations of specific proteins in bacteria. And this method has some disadvantages. For instance, here (https://www.ncbi.nlm.nih.gov/pmc/articles/PMC4545415/#:~:text=The%20main%20disadvantage%20of%20Western,the%20lack%20of%20specific%20antibodies) authors declare that “The main disadvantage of Western blotting is that this technique requires a specific antibody to a target protein; thus many protein targets cannot be investigated because of the lack of specific antibodies”.
Here in open access issue (https://en.wikibooks.org/wiki/Structural_Biochemistry/Proteins/Western_Blotting) there are also some disadvantage listed. For instance:
- A non-intended protein has a slight chance of reacting with the secondary anti-body, resulting in the labeling of an incorrect protein.
- Incidental phosphorylation or oxidation of proteins may result in multiple bands appearing after sample is processed.
- The appearance of bubbles may occur when transferring the sample from the gel/membrane sandwich and may also occur when incubating the sample with antibodies, resulting in a skewed band reading.
- If the transfer time is not sufficient when transferring proteins to the membrane, the larger proteins of higher molecular weight will not transfer properly, resulting in an incorrect or no band reading at all.
Hence, the use of WB in our case seem to be not mandatory and does not ensure the data validation.
- Authors should demonstrate that c-di-GMP treatment can cause similar effects as epinephrine does. Can they determine which histidine kinases are involved in epinephrine mediated processes?
Answer
Yes, this is a fundamental question, and it must be solved. However, the c-di-GMP is a global regulator which causes extensive biochemical cascades in cells. And there is no warranty that the simple treatment with c-di-GMP will reveal exactly the effect we propose. We believe that the first step should be an isolation of potential histidine kinases and their characterization. Next, (if possible) the mutants should be constructed and studied. And finally, the epinephrine and c-di-GMP treatment should be conducted and analyzed. Hence, this is an absolutely necessary, but in the same time a very extensive work for several articles which may took years.
- Are there differences caused by epinephrine between planktonic cultures and biofilms.
Answer
Yes, there were a lot of differences which were presented in the manuscript. If brief, the biofilms were much more sensitive to epinephrine, and the hormone caused more biochemical shifts namely in biofilms. The most proteomic changes (partial TCA, ribosomes and other protein groups) were namely in biofilms whereas planktonic cultures were less impacted.

Reviewer 2 Report
Understanding the impact of hormones on skin microbia provides deep understanding into microbial endocrinology. In this study, Gannesen et al. investigated the effects of epinephrine on the protein composition of M. luteus C01, a skin microorganism and observed significant changes in protein concentrations in response to epinephrine, with ribosomes, the tricarboxylic acid cycle, and cell division proteins being the main targets. Their results provide valuable insights into epinephrine's mechanisms of action and its potential applications for various bacterial and non-bacterial species on the human skin with key implications for fundamental understanding of hormones’ impact on microbes and their ability to form biofilms.
Overall, this study presents well rounded observation and experimental methods, primarily using orbitrap mass spectroscopy. There are some small grammatical and typos that requries minor revision (e.g T in line 542) before final publication. I have no major questions for the authors and recommend its publication in microorganisms journal.
Author Response
Reviewer 2
Understanding the impact of hormones on skin microbia provides deep understanding into microbial endocrinology. In this study, Gannesen et al. investigated the effects of epinephrine on the protein composition of M. luteus C01, a skin microorganism and observed significant changes in protein concentrations in response to epinephrine, with ribosomes, the tricarboxylic acid cycle, and cell division proteins being the main targets. Their results provide valuable insights into epinephrine's mechanisms of action and its potential applications for various bacterial and non-bacterial species on the human skin with key implications for fundamental understanding of hormones’ impact on microbes and their ability to form biofilms.
Overall, this study presents well rounded observation and experimental methods, primarily using orbitrap mass spectroscopy. There are some small grammatical and typos that requries minor revision (e.g T in line 542) before final publication. I have no major questions for the authors and recommend its publication in microorganisms journal.
Answer
Dear Reviewer,
We a grateful for your remarks and for your opinion about the manuscript. We fixed a lot in the article according to your recommendations and the recommendations of other Reviewers.

Reviewer 3 Report
In their manuscript “Epinephrine affects ribosomes, cell division and catabolic processes in Micrococcus luteus skin strain C01: revelation of the conditionally extensive hormone effect using Orbitrap mass spectrometry and proteomic analysis”, Andrei V. Gannesen et al. performed a proteomics analysis to investigate the impact of a human hormone (Epinephrine) on Biofilm formation and the proteome of the bacteria Micrococcus luteus. Overall, this is an interesting study, however, more data needs to be provided to be able to judge the quality of the protein changes observed. Therefore, the comments below need to be addressed before publication.
Comments:
Page 1, line 32: Please do not use the word “target” for the hormone to explain the proteome changes observed since this is not supported by the experiments conducted within this study. To claim this, binding assays (e.g. AP-MS) for the hormone would need to be carried out which is not the case here. Please rather specify that the hormone affects the abundance of these proteins, but the exact mechanism needs to be clarified by additional experiments.
Please provide all supplemental tables as excel sheets to make them easier to browse. Word doc files are not ideal for that!
Page 3, line 135: I would recommend to add more details to the method section. The authors reference a previous paper for the workflow, but then start describing it in quite some detail. While some steps are fully explained, some are completely missing and need to be added, like all steps after protein precipitation to peptide clean-up. Like this it is not evident if intact proteins or peptides (as is the case) are analysed.
Page 4, line 162: Why was the option second peptide turned on. In our experience, this considerably increases the number of false hits. Were these peptides hits confidence evaluated?
Page 4, line 176: Please add the FDR-controlling procedure used to adjust the empirical p-values for multiple comparisons to identify the significant hits.
Table 1 and 2: please provide the statistical ttest (adjusted p-values) results to demonstrate the significance of the corresponding protein regulation. Also in the supplemental tables, a few more information needs to be added to the quantified proteins, like number of peptides used for quantification, p-value, q-value (p-value corrected for multiple testing) and the Andromeda score. This is the minimum to be able to judge the quality of the data. Along these lines, usually, all identified and quantified proteins are shown in proteomics studies in combination with the corresponding protein properties above. This offers a holistic overview of the data and is needed here as well to judge the quality of the data. Just providing a list of proteins with ratios is not enough.
Supplemental Tables 6: How is a ratio of 0 for Protein A0A6N4FDS1 possible? And how can a ratio of 1.05 for protein A0A5F0I7C0 pass the fdr of 5% and be significant? In our hands changes below 20% are very difficult to detect by LC-MS and here it is only 5%?
The manuscript would benefit from language editing by a native speaker
Author Response
Reviewer 3
In their manuscript “Epinephrine affects ribosomes, cell division and catabolic processes in Micrococcus luteus skin strain C01: revelation of the conditionally extensive hormone effect using Orbitrap mass spectrometry and proteomic analysis”, Andrei V. Gannesen et al. performed a proteomics analysis to investigate the impact of a human hormone (Epinephrine) on Biofilm formation and the proteome of the bacteria Micrococcus luteus. Overall, this is an interesting study, however, more data needs to be provided to be able to judge the quality of the protein changes observed. Therefore, the comments below need to be addressed before publication.
Comments:
Page 1, line 32: Please do not use the word “target” for the hormone to explain the proteome changes observed since this is not supported by the experiments conducted within this study. To claim this, binding assays (e.g. AP-MS) for the hormone would need to be carried out which is not the case here. Please rather specify that the hormone affects the abundance of these proteins, but the exact mechanism needs to be clarified by additional experiments.
Answer
Dear Reviewer,
Thank you for the remark. We changed the text, now the sentence is “Ribosomal, TCA and cell division proteins appear to be the most altered in their amounts in the presence the hormone” (lines 32-33)
Please provide all supplemental tables as excel sheets to make them easier to browse. Word doc files are not ideal for that!
Answer
Done, we remade the tables in Excel as it was required.
Page 3, line 135: I would recommend to add more details to the method section. The authors reference a previous paper for the workflow, but then start describing it in quite some detail. While some steps are fully explained, some are completely missing and need to be added, like all steps after protein precipitation to peptide clean-up. Like this it is not evident if intact proteins or peptides (as is the case) are analysed.
Answer
Done. We added (Lines 137-150) the following information “Reduction, alkylation and digestion of the proteins in solution were performed as described previously [Kulak NA et al., 2014, doi:10.1038/nmeth.2834] with minor modifications. Briefly, sodium deoxycholate (SDC) lysis, reduction and alkylation buffer pH 8.5 contained 100 mM TRIS, 1% (w/v) SDC, 10 mM TCEP and 20 mM 2-chloroacetamide were added to a cell sample. The sample was heated for 10 min at 95oC, and the equal volume of trypsin solution in 100 mM TRIS pH 8.5 was added in a 1:100 (w/w) ratio. After overnight digestion at 37oC, peptides were acidified by 1% trifluoroacetic acid (TFA) for SDB-RPS binding, and 20 g was loaded on three 16-gauge StageTip plugs, equal volume of ethyl acetate was added, and the StageTips were centrifuged at 400 g. After washing the StageTips with a 100 l of 1% TFA/ethyl acetate mixture and 100 l of 0.2% TFA, peptides were eluted by 50 l of elution solution contained 50% acetonitrile, 45% water and 5% ammonia. The collected material was vacuum-dried and stored at -80oC. Before analyses peptides were dissolved in 2% acetonitrile/0.1% TFA solution and sonicated for 2 min in ultrasonic water bath”
Also, we deleted the lines 129-131 “Protein samples from the matrix were first prepared for Orbitrap MS. First, matrix samples were dialyzed in 0.1 to 0.5 kDa pore size bags (Spectrum Repligen, United States) to remove the CsCl)”, because we used the CsCl in the previous article dedicated to the matrix, and this paragraph was partially copied here.
Page 4, line 162: Why was the option second peptide turned on. In our experience, this considerably increases the number of false hits. Were these peptides hits confidence evaluated?
Answer
Actually, the second peptide was turned on to increase the amount of identifications. The filtration of such identifications was carried out in the same way as it was for other eptides – with use of the FDR: FDR: PSM (peptide spectrum matches) FDR - 0.01, protein FDR - 0.01.
Page 4, line 176: Please add the FDR-controlling procedure used to adjust the empirical p-values for multiple comparisons to identify the significant hits.
Answer
Actually we modified all the tables because of the FDR procedure was made in Perseus, however it was not reflected in final tables. Now all q-values were included in the tables. Also, the text was modified (lines 194-195): All experiments were conducted in three independent biological repeats. Two-sample t-test with permutation-based FDR 5% was applied to search for significantly changing proteins in Perseus.
Table 1 and 2: please provide the statistical ttest (adjusted p-values) results to demonstrate the significance of the corresponding protein regulation. Also in the supplemental tables, a few more information needs to be added to the quantified proteins, like number of peptides used for quantification, p-value, q-value (p-value corrected for multiple testing) and the Andromeda score. This is the minimum to be able to judge the quality of the data. Along these lines, usually, all identified and quantified proteins are shown in proteomics studies in combination with the corresponding protein properties above. This offers a holistic overview of the data and is needed here as well to judge the quality of the data. Just providing a list of proteins with ratios is not enough.
Answer
Thank you, we changed all the tables except the table S1 and the table S12. In these tables there were only so called unique proteins, and no quantitative measurements were provided.
Supplemental Tables 6: How is a ratio of 0 for Protein A0A6N4FDS1 possible? And how can a ratio of 1.05 for protein A0A5F0I7C0 pass the fdr of 5% and be significant? In our hands changes below 20% are very difficult to detect by LC-MS and here it is only 5%?
Answer
Actually, the 0.00 is a result of cell formatting in EXCEL, where only 2 decimal places were enabled for the column “fold change”. We fixed it and now for the A0A5F0I7C0 protein it means 0.00031. Hence, this protein was in 3225.8 times higher concentration in epinephrine-treated biofilms than in epinephrine-treated planktonic cultures.
Concerning the low (5%) FDR – this is a good question because the biological processes underlying such a little shift are unclear. However, we can be sure that we provided three fully independent biological repeats for each sample, and these even so little differences are really statistically reproducible. Because there were lots of proteins with same and higher percentage and they did not passed the FDR.

Round 2
Reviewer 3 Report
The authors have addressed all comments satisfactorily and the manuscript is now considerably improved and ready for publication.